# Preclinical Detection of Early Glomerular Injury in Children with Kidney Diseases—Independently of Usual Markers of Kidney Impairment and Inflammation

**DOI:** 10.3390/ijms25179320

**Published:** 2024-08-28

**Authors:** Heidrun Rhode, Baerbel Tautkus, Friederike Weigel, Julia Schitke, Oliver Metzing, Jan Boeckhaus, Wieland Kiess, Oliver Gross, Axel Dost, Ulrike John-Kroegel

**Affiliations:** 1Jena University Hospital, Institute of Biochemistry I, Nonnenplan 2-4, 07743 Jena, Germany; 2Jena University Hospital, Department of Pediatrics and Adolescent Medicine, Am Klinikum 1, 07747 Jena, Germany; 3Clinics for Nephrology and Rheumatology, University Medical Center Göttingen, Robert-Koch-Str. 40, 37075 Göttingen, Germany; jan.boeckhaus@med.uni-goettingen.de (J.B.); gross.oliver@med.uni-goettingen.de (O.G.); 4Hospital for Children and Adolescents, University of Leipzig, Liebigstr. 20a, 04103 Leipzig, Germany

**Keywords:** biomarker discovery, predictive analysis, rare genetic disorders, orphan disease, glomerulonephritis, polycystic kidney disease, congenital anomalies of kidneys and urogenital tract, pre-clinical

## Abstract

Glomerular kidney diseases typically begin insidiously and can progress to end stage kidney failure. Early onset of therapy can slow down disease progression. Early diagnosis is required to ensure such timely therapy. The goal of our study was to evaluate protein biomarkers (BMs) for common nephropathies that have been described for children with Alport syndrome. Nineteen candidate BMs were determined by commercial ELISA in children with congenital anomalies of the kidneys and urogenital tract, inflammatory kidney injury, or diabetes mellitus. It is particularly essential to search for kidney disease BMs in children because they are a crucial target group that likely exhibits early disease stages and in which misleading diseases unrelated to the kidney are rare. Only minor differences in blood between affected individuals and controls were found. However, in urine, several biomarker candidates alone or in combination seemed to be promising indicators of renal injury in early disease stages. The BMs of highest sensitivity and specificity were collagen type XIII, hyaluronan-binding protein 2, and complement C4-binding protein. These proteins are unrelated to inflammation markers or to risk factors for and signs of renal failure. In conclusion, our study evaluated several strong candidates for screening for early stages of kidney diseases and can help to establish early nephroprotective regimens.

## 1. Introduction

Chronic kidney disease (CKD) global prevalence is ~13.4%, and global adult CKD is increasing [1]. CKD starts without clinical symptoms despite early renal damage. CKD often progresses to end stage kidney disease (ESKD). Thus, to optimally delay CKD progression, therapy is most beneficial to asymptomatic patients, i.e., before onset of proteinuria [2,3,4,5]. Therefore, most nephropathy patients would benefit from early diagnostic and prognostic BMs. 

The most frequent renal disease is diabetic kidney disease (DKD). This disease is caused mainly by hyperglycemia induced glomerular alterations, including GBM thickening and podocyte injury [6,7]. For identifying DKD, current guidelines recommend tests for microalbuminuria. However, microalbuminuria already implies glomerular malfunction [8,9,10]. Novel BMs of DKD are most likely among the products of early disease processes (e.g., focal structure damage, focal immune reactions, and inflammation) [11]. Many proposed BMs for advanced CKD and DKD derive from immune reactions and are possibly related to disease progression [12]. These proposed BMs in the blood include acute phase reactants (APRs) [10], constituents from cells and the extracellular matrix [11,13,14,15,16], including tubular segments, metabolites [15,17], (exosomal) micro-RNAs [18,19,20,21,22], exosomes [23], and urinary extracellular vesicles (uEV) [24,25]. They also include typical chemical modifications [26], epigenetic imprinting [20,27], endothelial cell injury markers [26], products of acute kidney injury (e.g., cystatin C), and products of fibrosis, like TGF-β1 [28,29]. Several tubular BMs are associated with GFR decline in childhood-onset type 1 DM (T1DM) [30] and T2DM [31]. Moreover, differentially expressed genes (DEGs) [32], focally altered proteins [33], and urinary peptide signatures were used to specify the deviation of particular BM proteins [34]. Most of these BMs require specialized techniques that are often unavailable. Current tools to diagnose CKD before onset of albuminuria include initial hyperfiltration [35], metabolites [36], constituents of uEV [24,37], and proteins [38], including signs of early immune reactions. In addition, urinary vascular endothelial growth factor (VEGF-A), angiotensinogen (AGT), transferrin [39,40], and α1-acid glycoprotein (a1AGP) [41] were higher in normoalbuminuric children with T1DM than in controls and correlated with the urine albumin/creatinine ratio (UACR). 

Obesity-related glomerulopathy (ORG) is associated with CKD development [42,43]. Its pathophysiology includes hemodynamic changes, activation of the renin-angiotensin-aldosteron system (RAAS), endocrine and metabolic alterations, inflammation, and oxidative stress [42]. There are several proposed new BMs of early kidney injury in ORG, e.g., tubular BMs of acute kidney injury as well as slit diaphragm constituents [42]. 

In children, CKD is commonly due to congenital disorders and glomerular diseases [44]. Delaying CKD progression requires early, pre-symptomatic diagnosis. Therefore, BMs of early, pre-symptomatic disease and monitoring disease progression need to be sensitive enough to distinguish between affected and unaffected children [45,46,47]. 

In our previous work, we took use of animal models for Alport syndrome (AS), an inherited type IV collagen glomerulonephropathy. Type IV collagen is preserved in all mammalian species. Therefore, it was possible to use Alport mammalian models to study pathogenesis and therapy [48], to study glomerular injury, and to search for early BMs [49]. 

Because most kidney alterations start at or affect the glomerulus, similar mechanisms are likely at early stages of most CKD. Thus, according to our hypothesis, BMs can be unspecific to disease but specific to the injured glomerulus. For that reason, our study evaluated early protein biomarkers for common nephropathies. The goal was to determine the ability of our BMs to distinguish healthy individuals from those with different nephropathies in a way suitable for point-of-care early screening.

## 2. Results

In our previous studies [49,50], mammalian models of Alport syndrome and children with Alport syndrome were screened initially to identify generic markers for glomerular disease. In our current study, these markers are now being tested in patients with various kidney diseases (including CAKUT, cystic disease, and diabetes) to see whether they represent useful BMs for other forms of kidney disease (Figure 1). In the interest of clarity, our multi-step program was divided into a discovery phase and a proof-of-concept phase. The **discovery phase** (see Figure 1) included a mass spectrometric search for pre-clinical BMs in two animal models and the verification of a panel of these BMs in children with AS and similar diseases [49,50]. These BMs occur in body fluids and originate from injured glomerular structures. 

In the **proof-of-concept phase**, we analyzed the same panel of BMs to determine whether they indicated early kidney injury in renal disorders other than AS (Figure 2). Again, we performed this determination in samples from children because it is in children that renal alterations characteristic of early or even of a pre-clinical stage of CKD can be expected.

### 2.1. Biomarkers Previously Identified in Early AS Discriminated between Healthy Children and Those with Various Other Nephropathies

Figure 3 shows four individual BMs with the greatest discriminatory power, and Table 1 summarizes all the BMs that were considerably different between patients and healthy controls (visualization in Appendix A). In serum, four BMs discriminated between people with nephropathies and healthy controls. These were procollagen type I carboxy-terminal propeptide (PICP), TGF-β1, vitronectin (VTN), and CRP. VTN and CRP were indicative in patients at risk of pre-clinical DKD. Nevertheless, the median VTN was only slightly higher in children with nephropathies than in controls, and the group ranges overlapped. However, CRP of <1 mg/L, i.e., below commonly applied concentration ranges, distinguished between groups. 

In contrast to the situation in serum, in urine, 17 BMs were significantly different, mostly more abundant in patients than in controls. These 17 BMs were able to identify more than 40 times when patients had significantly higher values than controls. Among these BMs, three identified most disease groups. These three were collagen type XIII (ColXIII), hyaluronan-binding protein 2 (HABP2), and complement C4-binding protein (C4BP). They were also able to identify children in groups A and B presumed to have pre-clinical stage KD. These individuals were those with obesity, metabolic syndrome, or T2DM and T1DM. 

Because several urinary BMs correlated with each other (Appendix A), we checked 26 products of positively correlated urinary BMs for their ability to enhance discrimination between patients and controls. 

Several combinations indeed discriminated superiorly between patients and controls than single marker concentrations (Figure 4, Table 2), as in AS [49]. Altogether, there were 70 instances of renal injury in which we identified significantly higher values than in controls. In practice, it was possible to detect all nephropathies using at least one of these combinations. For example, we were able, using several combined BMs, to detect early KD in children with obesity, metabolic syndrome, hypertension, and T2DM (group A). Ten combinations were significantly greater than controls in T1DM patients (group B), and 19 in ADPKD/ARPKD patients (group E).

### 2.2. Several Individual BMs and Combinations of BMs Showed Valuable Diagnostic Scores

The AUCs were above ~0.700 for ColXIII, HABP2, and C4BP, as well as combinations thereof (e.g., Table 3, Figure 5). These BMs also had reliable quality characteristics (sensitivity, false positive rate), either alone or in combination. This held true both for several sub-groups analyzed separately (Appendix A) and for all nephropathy patients grouped together (Appendix A). In post-infection GN, IgAN, and IgAV (group G), some APRs (CRP, AGT, fibrinogen, complement factors I and H (CFI, CFH)) also produced high accuracy (Appendix A).

### 2.3. Correlation of Our BMs with Clinical Parameters

In this part of our study, small sample sizes prevented complete analyses for some BMs. Nevertheless, in the **control group**, no BM candidate correlated with BMI-SDS (Appendix A). Only serum AGT and C1q correlated with age (Appendix A). The only BM that correlated (inversely) with diastolic blood pressure was serum ADP (Appendix A). There was no need for age- or BMI-related cutoffs for any of our urinary BMs. In **group A (obesity, metabolic syndrome, hypertension and T2DM)**, UACR did not correlate with any of our BM candidates. There were, however, notable relationships of urinary AGT, gelsolin (GS), VTN, and complement C9 (C9) with concurrent blood glucose levels (Appendix A). There were no such relationships with HbA1c, BMI, and CRP, however (Appendix A). Systolic blood pressure was consistently correlated with urinary ADP (Appendix A), and diastolic blood pressure with serum PICP, TGF-β1, and urine C9. There was also no correlation of albuminuria in **group B (T1DM)** with any BM candidates. The associations seemingly present for urinary a1AGP with PICP, serum CRP with UACR, and of a1AGP with blood glucose were all due to single data points. (Appendix A). There was also no relationship between any BM with HbA1c, BMI, blood pressure (Appendix A), or with disease duration (Appendix A). **Groups C to G** all had few members. We therefore combined all pediatric patients diagnosed with nephropathies for correlation analysis. Combined, UACR and serum creatinine did not correlate with any of our BMs except urinary lumican (LUM), which was related to microalbuminuria (Appendix A). There was no correlation of any BM with eGFR or cGFR (Appendix A), and only a few samples produced a weak inverse correlation of cGFR with serum CFH. No BM was related to serum or urine CRP, i.e., to an inflammation degree, which is absent or subclinical, applying the current threshold (Appendix A). In nephropathies, no BM was associated with blood pressure (Appendix A).

### 2.4. Sex Differences

There were no gender-dependent differences in individual characteristics or main clinical data in our control group (Table 1). Moreover, in all groups of sufficient sample size, there were no consistent differences in BM concentrations (Appendix A) between males and females. There were thus no sex-specific cutoffs.

## 3. Discussion

The most promising biomarker candidates in the urine with the highest sensitivity and specificity for early disease stages were collagen type XIII, hyaluronan-binding protein 2, and complement C4-binding protein. We verified these pre-clinical BMs in both pre-clinical and in early stages of the different nephropathy types, acquired and inherited, provoked by differing pathogenetic dysfunctions. 

Urinary ColXIII, HABP2, and C4BP are BMs of glomerular origin. ColXIII is a glomerular endothelial membrane protein [51], HABP2 a GBM constituent, and C4BP a regulatory complement component. The last two are produced by podocytes [52]. Thus, these BMs are probably specific for early glomerular injury, independently of the renal impairment indicators usually applied. 

In contrast, serum CRP was not greater in pre-clinical AS than in controls, supporting the lack of overt systemic inflammation pre-clinically [49]. In this study, most juvenile patients were also in pre-clinical nephropathy, although few were at a later stage. It is notable that, in both cases, CRP inflammation marker concentrations lay within the currently accepted normal range.

Children with **obesity, metabolic syndrome, hypertension, and T2DM (group A)** are at high risk of developing kidney impairment [53,54]. In our study, these children were still in the pre-clinical CKD stage, including a normal GFR (Table 1 and Table 4). Nevertheless, patients had significantly higher concentrations than healthy controls of CRP in serum and urine, of ColXIII, HABP2, and C4BP only in urine, and of VTN only in serum. In conclusion, these BMs were highly diagnostic either alone or in combination for children of group A.

Circulating, liver-derived AGT passes through damaged glomerular filtration barriers, but it is also released from the proximal tubules into the urine [57]. In mice, AGT is raised at high levels of advanced glycation end-products [58], and, matching our results, high glucose levels [57]. Urinary AGT is increased by hypertension, manifested CKD, IgAN, and DM [59]. We were unable to confirm that AGT indicates kidney injury in DKD before albuminuria [60]. Thus, urinary AGT is potentially a BM for intra-renal RAAS status and for kidney impairment of various origins. However, none of our BMs correlated with BMI, though there were some relationships with blood pressure. We were unable to verify in our pediatric patients the supposed association of high urinary ADP with HbA1c and UACR in T2DM [58,61].

The correlations of urinary AGT, GS, VTN, and C9 with concurrent blood glucose were determined by blood sugar values > 7.5 mM. In consequence, they are possibly related to High Glucose-Induced podocyte injury [62].

The group with **T1DM (group B)** had normal parameters (serum creatinine, UACR); however, UACR and blood pressure were slightly higher and cGFR was slightly lower than in controls (Table 4). We were unable to confirm the increasing relationship of a1AGP with UACR [41]. Nevertheless, serum a1AGP correlated with blood glucose, but this was due to just two hyperglycemic values. However, as in group A, there was no correlation of any of our urinary BMs with HbA1c or BMI. There was likewise no correlation with blood pressure or with the duration of disease. Surprisingly, no urinary BMs correlated with blood glucose, as in group A. The lack of a correlation was also true of AGT, although such a correlation has been reported [40]. In this group, combinations of urinary ADP, PICP, HABP2, and TGF-β1 had the highest diagnostic scores for pre-clinical renal alterations.

For all DKD, ORG, and metabolic-syndrome-associated KD, the complement system is crucial for CKD development and progression [43]. In contrast, in our study, most complement components were not indicative in pre-clinical pediatric patients. The only exceptions were urinary C4BP (in group A) and serum C9 (in group B). 

Most pediatric CKD **(nephropathies, groups C to G)** are inherited and involve renal mass loss (C, E), vesico-urethral reflux (D), or inflammatory nephropathies (G). Elevated glomerular filtration-pressure, RAAS-activation, and inflammation affect the glomeruli and other parts of the nephrons. In consequence, affected children are at increased risk of glomerular injury and impaired kidney function. All CKD, derived from obstructive uropathy to glomerulonephritis, have overlapping pro-inflammatory and pro-fibrotic pathways, whatever the primary disease [63]. These diseases are usually diagnosed from family history, genetics, ultrasound scans [64], and biopsy. Our novel and early BMs are potentially able, however, to help identify injury indicating signals earlier than by the means usually applied. Moreover, such BMs are also possible aids in the development of new therapies for delaying renal failure [65], as in AS [66]. We assume that raised BM levels indicate early renal impairment because most of our patients showed no or mild renal impairment and only slightly higher-than-normal blood pressure (Table 1). In group F (nephronophthisis, a primary tubulointestinal injury), our BMs appear, as expected, practically no different from controls.

Our promising BMs indicate renal (glomerular) injury. Their diagnostic scores are slightly different, however, in our sub-groups, and slightly lower than in AS [49]. In groups C and D, urinary HABP2 had the highest score. In group H, urinary HABP2 also had the highest score, but in combination with one of the other BMs (fibrinogen, C4BP, or ColXIII). We also found CFH and CFI to be moderately indicative in groups F and G. No one has yet analyzed plasma CFH as an early diagnostic BM, but an attempt was made to use it to differentiate between children with CAKUT CKD and those with non-CAKUT CKD [67].

Gelsolin (GS) is one of a family of actin-interacting proteins mainly implicated in cytoskeletal structure remodeling [68]. In the kidney, it is mainly expressed in the tubular system [69]. Plasma GS has actin clearing as well as numerous immune modulatory activities. Consequently, GS is affected by a wide variety of injuries, including CKD, and its abundance is then usually reduced. Additionally, its levels correlate with disease progress and mortality [68,70,71,72]. GS is thus probably a general prognostic health marker [71]. Even urinary GS is a rather non-specific marker of diverse serious disturbances [73]. Nevertheless, GS is a BM more abundant in AS-mice at the earliest stage [50], in AS-dogs, and in children with AS [49] than in controls. GS also has usable diagnostic scores in groups F and G, mainly in combination with other BMs.

Because IgAN and IgAV are inflammatory diseases, several APRs (including complement regulators) were, as expected, more abundant than in controls. All BMs identified in AS were clearly elevated in these patients with high diagnostic scores. This is a strong validation of our biomarkers. We confirmed that urinary AGT is more abundant in IgAN disease than in healthy controls [74]. 

The relationship of urinary lumican and UACR was determined by some microalbuminuric values. There was a weak and opposing correlation of CFH with eGFR and cGFR, and the correlation with serum CRP was determined by a single data point from an obese subject. There is no relation with BP. In children, there is also no correlation of C1q with GFR in blood nor in urine, despite one in adults [75].

In AS, we were unable to identify as BMs several proposed markers of glomerular injury [49,76]. Therefore, we did not examine their validity. These proposed markers include typical constituents of the slit diaphragm, the podocytes themselves, or filtrated plasma proteins combined with several tubular products [76].

Interestingly, our study found CRP in urine to be more than three orders of magnitude lower than in serum. In patients with nephropathies (group C to G), there was no correlation between urine and serum CRP concentrations and only a weak one in groups A and B. However, both urine and serum CRP had discriminatory power in our groups with inflammatory diseases (groups A and G). Nevertheless, we do not recommend CRP for screening because it is not specific to KD. This is particularly likely to be problematic in children because children frequently suffer from other subclinical and clinically overt infections.

As a limitation of our study, our sample sizes were low because there were few routine visits from patients during our work due to the coronavirus pandemic. Nevertheless, even with the limited sample sizes, there were strong significant differences between patients and controls. Several blood-serum parameters were able to discriminate between patients and controls. Due to its higher discriminatory power and non-invasiveness, however, urine is clinically preferable for practical reasons. This is particularly true for pediatric practice. Moreover, urine contains more BMs suitable as diagnostic tools.

The goal of our biomarker approach was to identify children at early stages of CKD, which might translate to a long-term clinical benefit: a further delay of progression of CKD by early diagnosis and therapy. The potential of early diagnosis of CKD in children to delay ESKD and improve life-expectancy has been recently discussed in *Clinical Practice Guideline for Microhematuria in Children and Young Adults* [77].

## 4. Materials and Methods

### 4.1. Patients

We measured BMs in samples from children with various CKDs diagnosed clinically, by family history, ultrasound, histology, genetic analysis, and established renal parameters. Patients with related diseases were grouped for statistical analysis:
(A)obesity, metabolic syndrome, manifested T2DM, and arterial hypertension (B)T1DM(C)functional solitary kidney (renal agenesis, multicystic dysplastic kidney, and unilateral hypoplasia without renal function) (D)congenital anomalies of the kidney and urinary tract (ureteropelvic stenosis, duplex kidneys, vesico-ureteral reflux, and lower urinary tract obstruction) (E)autosomal dominant and recessive polycystic KDs (ADPKD, ARPKD) (F)hereditary ciliopathy (juvenile nephronophthisis) (G)glomerular diseases (IgAN, IgA-vasculitis (IgAV), post-infectious glomerulonephritis (GN)).

### 4.2. Samples

We used three sample sets from humans. These samples were transported and stored below −80 °C.

**Set 1** came from pediatric patient groups (see above). We determined KD stages according to KDOQI-guidelines [78]. Most patients (131/147, 89.1%) were at a pre-clinical stage based on their GFR and albuminuria, i.e., stage G1A1. In group A, all children had eGFR > 90 mL/1.73 m^2^ and UACR < 20 mg/g creatinine. In group B, only one child (1/47, 2%) had eGFR < 60 mL/1.73 m^2^ (G3a/A1), and one other UACR > 300 mg/g (G1A3). All others were in stage G1A1. None of the people in the nephropathy groups (groups C to G) had eGFR < 100 mL/1.73 m^2^. However, twelve patients (12/80, 15%) had microalbuminuria (G1A2), and two children (2/80, 2.5%) with polycystic KDs had UACR > 300 mg/g (G1A3).

**Set 2** came from children without diabetes or nephropathy as a control group at the Department of Pediatrics of the University Hospital in Jena. 

Sample sets 1 and 2 were acquired applying standard procedures (09/2020 to 09/2021, ethical approval UHJ: 2020-1800) and included blood and spot urine.

**Set 3**, spot urines, came from an independent cohort of pediatric control samples from the Leipzig Medical Biobank (2014, Leipzig Research Center for Civilization Diseases, LIFE-child). This set is identical with set 3 in our AS research 49 and allowed tracking over 7 years (Table 4).

### 4.3. Evaluation by Immune Assays

All measurements were in duplicate. Before measuring the sample values, we determined the measurable range and dilutional linearity of each assay. (ELISA test kits applied in Appendix A). All concentrations of BMs in urine were normalized by the respective urinary creatinine concentration.

### 4.4. Statistical Analysis of ELISA Results

We analyzed the data using SPSS™ (v. 27). The data were non-normally distributed in all patient sub-groups (Shapiro Wilk Test). Therefore, we tested group similarity by Kruskal–Wallis tests and post hoc pairwise comparison by Mann–Whitney U test. Statistical significance was set at *p* < 0.05 (95% confidence interval, CI). To study BMs in combination, we correlated the immune reactivity of BMs with each other and multiplied positively correlated proteins. We used correlations with several clinical measures and receiver operating characteristic curve (ROC) analyses to assess the diagnostic values of single and combined BMs. The interpretation followed a published protocol [79] with moderate accuracy, defined as area under the ROC curve (AUC) 0.7–0.9 and high accuracy as AUC > 0.9. We rate a specific BM as indicating disease when its mean concentration is significantly different from those of healthy controls. Its suitability for detecting a disease is also given when the lower bound of the 95% CI of its AUC is >0.500 and its misclassification rate is low.

## 5. Conclusions

We identified three BMs as valid for the heterogeneous nephropathies we studied. These three BMs (ColXIII, HABP2, C4BP), together with several others, reliably distinguish affected children from healthy controls. They are not related to commonly assessed risk factors for kidney injury such as BMI, BP, and long-term glycemic conditions (HbA1c). They are also not associated with usual renal failure indicators (e.g., GFR and serum creatinine) or correlated with established inflammation markers. Thus, these BMs indicate very early CKD by detecting non-specific processes of kidney injury. The origin of these BMs suggests these processes are particularly likely to be glomerular disturbances. Because their diagnostic score is slightly lower than under AS and nothing is currently known of their time courses, we strongly suggest using BM combinations for diagnostics. Our validation identifies combinations as superior to single parameters. Furthermore, besides the need for further studies using larger and more equal sample sizes per group, the time courses of all the BMs applied must be determined with enough patients to improve their applicability and diagnostic scores. Given such information and their diagnostic ability and specificity, our BMs are potentially very valuable screening parameters and require evaluation in clinical practice.

## Figures and Tables

**Figure 1 ijms-25-09320-f001:**
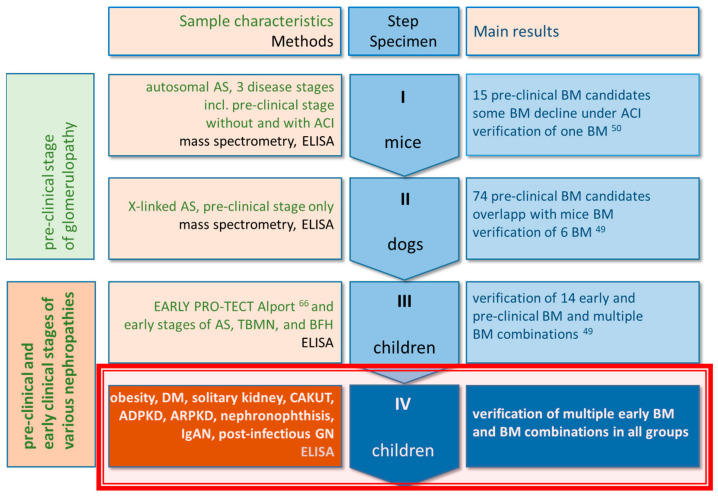
Scheme of our overall workflow including, all earlier and current research. The focus of our current study is the proof-of-concept step IV in children (red frame) with different kinds of CKD. Steps I (mice), II (dogs) and—in parts—III (children) have been published in [49,50]. Abbreviations: ACI, angiotensinogen converting enzyme inhibition; TBMN, thin basement membrane nephropathy; BFH, benign familial hematuria; DM, diabetes mellitus; CAKUT, congenital anomalies of kidney and urinary tract; ADPKD and ARPKD, autosomal dominant and recessive polycystic KDs; IgAN, IgA-nephropathy; GN, glomerulonephritis.

**Figure 2 ijms-25-09320-f002:**
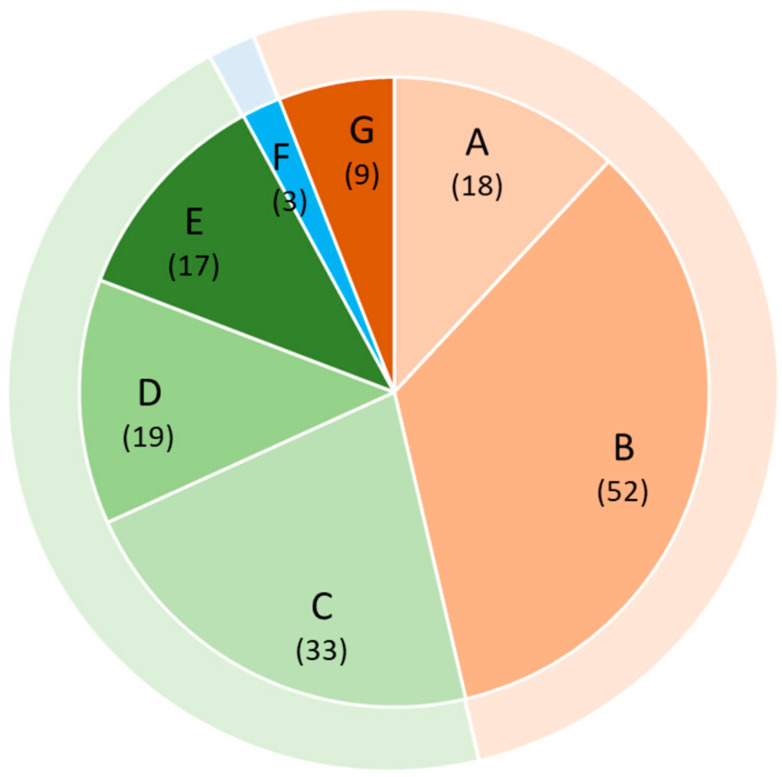
Visualization of patient categories and sample sizes. Patient groups: A: Obesity, metabolic syndrome, manifested T2DM, and arterial hypertension; B: T1DM; C: functional solitary kidney (renal agenesis, multicystic dysplastic kidney, and unilateral hypoplasia without renal function); D: congenital anomalies of kidney and urinary tract (including ureteropelvic stenosis, duplex kidneys, vesico-ureteral reflux, and lower urinary tract obstruction); E: autosomal dominant and recessive polycystic KDs (ADPKD, ARPKD); F: hereditary ciliopathy (juvenile nephronophthisis); G: glomerular diseases (IgA-nephropathy, IgA-vasculitis (IgAV), post-infectious glomerulonephritis (GN)). Outer ring: Orange: primary glomerulopathies; green: congenital nephropathies; blue: primary tubulointerstitial nephropathy. “()” indicate number of patients.

**Figure 3 ijms-25-09320-f003:**
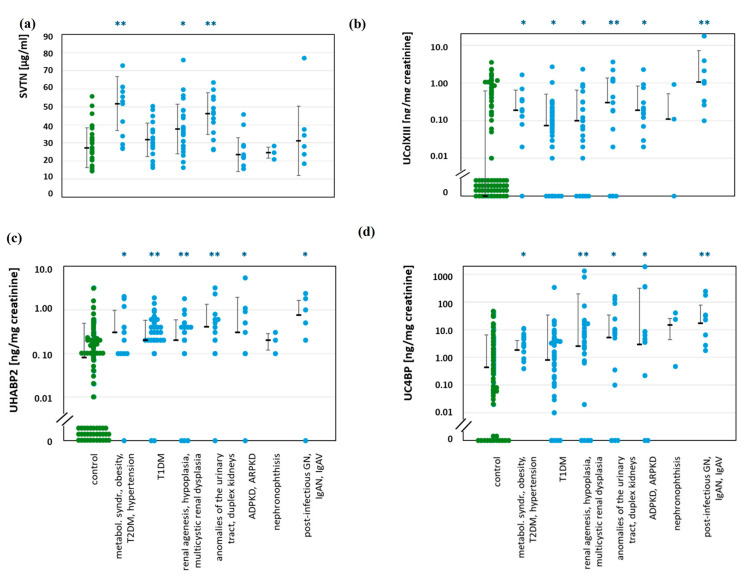
Examples of concentrations of individual BMs in serum and urine from patients with various nephropathies in comparison to healthy controls. Median, SD, and single data points are shown. Patient groups are identical in all charts, as given at the bottom. Significances: *p* < 0.05 (*); *p* > 0.01 (**); all other not significant. B to D: Values below the detection limit (=zero) are arranged close to the abscissa for visualization. All other significant differences are summarized in Table 2. (**a**): Vitronectin (VTN) in serum. (**b**): Collagen type XIII (ColXIII) in urine. (**c**): Hyaluronan binding protein 2 (HABP2) in urine. (**d**): Complement C4 binding protein (C4BP) in urine.

**Figure 4 ijms-25-09320-f004:**
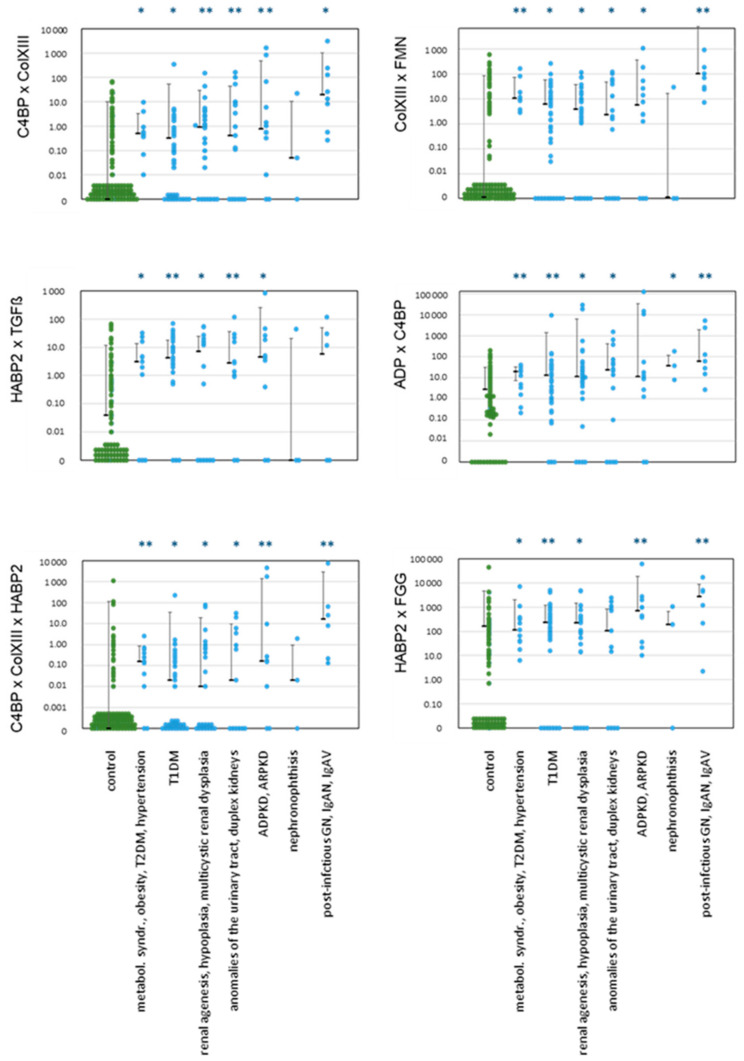
Examples of products of concentrations of BMs in urine from patients with various nephropathies in comparison to healthy controls. Median, SD, and single data points are shown. Patient groups are identical in all charts, as given at the bottom. Significances: *p* < 0.05 (*); *p* > 0.01 (**); all other not significant. Values below the detection limit (=zero) are arranged close to the abscissa for visualization. All other significant differences are summarized in Table 3. Abbreviations: collagen type XIII (ColXIII), complement C4 binding protein (C4BP), hyaluronan binding protein 2 (HABP2), formin 1 (FMN), adiponectin (ADP), fibrinogen gamma chain (FGG). All concentrations of BMs were normalized by urinary creatinine, like in Figure 3. These normalized values were multiplied as given on the *y*-axis.

**Figure 5 ijms-25-09320-f005:**
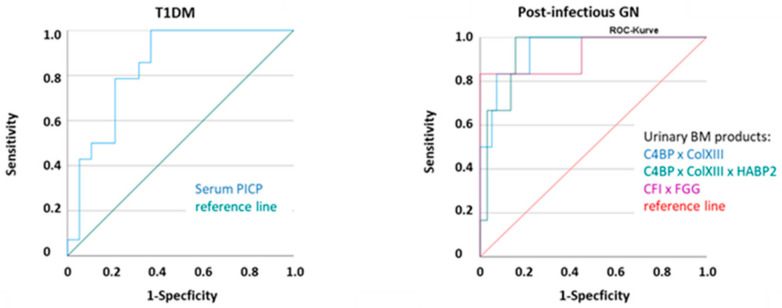
Examples of ROC curves of some BMs in patients with T1DM and post-infection glomerulonephritis, IgAN, and IgAV. All results, cutoffs, AUC, and sensitivity and specificity values are summarized in Appendix A. Abbreviations: collagen type XIII (ColXIII), complement C4 binding protein (C4BP), hyaluronan binding protein 2 (HABP2), fibrinogen gamma chain (FGG), complement factor I (CFI), procollagen type I carboxy-terminal propeptide (PICP). Products of BM concentrations were built as in Figure 4.

**Table 1 ijms-25-09320-t001:** Significance levels (*p*-values, *p*) of individual BM concentrations in serum and urine different in patients compared with healthy controls. n: number of patient values are given as minimum and maximum because all parameters in each group were determined on different numbers of samples. *p* < 0.05 (light grey) is considered significant and *p* < 0.01 (dark grey) highly significant according to Mann–Whitney U test. All other comparisons gave non-significant differences. We also included *p*-values of 0.05 to 0.10 to illustrate tendencies. Bold: favored BMs. Urinary concentrations were normalized by creatinine (c). Healthy controls: n = 11–25 for serum; n = 86–101 for urine. Abbreviations: adiponectin (ADP); angiotensinogen (AGT); collagen type XIII (ColXIII); α1-acid glycoprotein (a1AGP); hyaluronan binding protein 2 (HABP2); complement C4 binding protein (C4BP); complement factor H (CFH); complement factor I (CFI); fibrinogen gamma chain (FGG); gelsolin (GS); leucine rich glycoprotein 1 (LRGP1); vitronectin (VTN); complement component C1q (C1q); C-reactive protein (CRP); complement component C9 (C9); formin 1 (FMN); procollagen type I carboxy-terminal propeptide (PICP); transforming growth factor beta 1 (TGF-β); lumican (LUM).

Parameter	Concentration range	Obesity, Metabolic Syndrome, Hypertension, T2DM	T1DM	Renal Agenesis, Multicystic Renal Dysplasia	Anomalies of the Urinary Tract, Duplex Kidneys	ADPKD, ARPKD	Nephronophthisis	Post-Infectious GN, IgAN, IgAV
**Serum**								
n of patients		7–16	11–19	11–32	10–17	5–15	3	3–8
ADP	ng/mL							
a1AGP	mg/mL							
AGT	ng/mL							
ColXIII	ng/mL			0.066				
GS	µg/mL							
LRGP1	pg/mL							
HABP2	ng/mL					0.066		
**PICP**	ng/mL		<0.001		0.077			
TGFβ	ng/mL			0.049		0.047		0.068
**VTN**	µg/mL	0.003		0.032	0.001			
C9	ng/mL		0.056					
C4BP	ng/mL					0.044		
**CRP**	ng/mL	0.003						0.020
LUM	ng/mL							
FMN	pg/mL							
CFH	ng/mL							
CFI	ng/mL							
FGG	ng/mL				0.074		0.067	
C1q	ng/mL							
**Urine**								
n of patients		10–20	40–44	25–33	9–18	10–16	3	6–8
ADP	ng/mg c		0.013					
a1AGP	ng/mg c					0.064		0.003
AGT	ng/mg c			0.050		0.042		<0.001
**ColXIII**	ng/mg c	0.013	0.031	0.044	0.004	0.005		<0.001
GS	ng/mg c					0.016		0.007
LRGP1	ng/mg c							0.019
**HABP2**	ng/mg c	0.006	<0.001	<0.001	<0.001	0.012		0.020
PICP	pg/mg c		<0.001			0.003		0.002
TGFβ	pg/mg c		0.002					
VTN	ng/mg c			0.046		0.003		0.029
C9	pg/mg c						0.053	
**C4BP**	ng/mg c	0.009		0.002	0.015	0.028		<0.001
CRP	pg/mg c	<0.001				0.065		<0.001
CFH	ng/mg c			0.045		0.011		0.006
CFI	ng/mg c					0.020		<0.001
FMN	pg/mg c							0.002
LUM	ng/mg c							
FGG	ng/mg c					0.014		<0.001
C1q	ng/mg c			0.029	0.035			

**Table 2 ijms-25-09320-t002:** Significance levels (*p*-values, *p*) of combined BM concentrations in urine from patients compared with healthy controls. n: number of patient values are given as minimum and maximum because all parameters in each group were determined on different numbers of samples. *p* < 0.05 (light grey) is considered significant and *p* < 0.01 (dark grey) highly significant according to Mann–Whitney U test. All other comparisons gave non-significant differences. We also included *p*-values of 0.05 to 0.10 to illustrate tendencies. Bold: combinations with highest discriminatory power. Healthy controls: n = 84–99. Abbreviations: adiponectin (ADP); angiotensinogen (AGT); collagen type XIII (ColXIII); α1-acid glycoprotein (a1AGP); hyaluronan binding protein 2 (HABP2); complement C4 binding protein (C4BP); complement factor H (CFH); complement factor I (CFI); fibrinogen gamma chain (FGG); gelsolin (GS); leucine rich glycoprotein 1 (LRGP1); vitronectin (VTN); complement component C1q (C1q); C-reactive protein (CRP); complement component C9 (C9); formin 1 (FMN); procollagen type I carboxy-terminal propeptide (PICP); transforming growth factor beta 1 (TGF-β). Products of BM concentrations were built as in Figure 4.

Product of BMs in Urine	Obesity, Metabolic Syndrome, Hypertension, T2DM	T1DM	Renal Agenesis, Multicystic Renal Dysplasia	Anomalies of the Urinary Tract, Duplex Kidneys	ADPKD, ARPKD	Nephronophthisis	Post-Infectious GN, IgAN, IgAV
n of patients	10–13	41–43	25–30	9–17	10–13	3	5–8
ADP × PICP		<0.001			0.009		0.006
ADP × C9		0.011					
ADP × C4BP	0.030		0.004	0.044	0.068	0.026	<0.001
AGT × GS					0.007		0.001
AGT × C9		0.019			0.007		
**ColXIII × C4BP**	0.002	0.005	<0.001	0.010	0.002		<0.001
**ColXIII × FMN**	0.002	0.005	0.014	0.021	0.021		<0.001
GS × CFH			0.056		0.006		0.002
GS × C9					0.007		
GS × VTN					0.008		0.003
GS × CFI					0.001		0.001
LRGP1 × C1q			0.025	0.059			0.033
**HABP2 × TGFβ**	0.024	<0.001	0.056	0.002	0.005		
**HABP2 × FGG**	0.007	<0.001	0.006	0.084	<0.001		0.001
VTN × AGT		0.050	0.012		0.035		0.003
VTN × C9		0.025					
VTN × CFI					0.002		<0.001
C9 × CFH	0.082			0.099	0.024		
CRP × C1q	0.008			0.021			0.002
CFH × CFI				0.068	0.003		<0.001
CFH × FGG			0.096		0.002		<0.001
CFI × FGG					0.003		<0.001
CFI × a1AGP					0.035		<0.001
CFI × GS					0.053		0.026
FMN × FGG				0.017			<0.001
**ColXIII × C4BP × HABP2**	<0.001	0.006	0.016	0.015	<0.001		<0.001

**Table 3 ijms-25-09320-t003:** ROC analysis and comparison of urine BM values from patients with ADPKD and ARDKD (group E) with healthy controls. ROC analysis by SPSS (cf. Methods section). Only BMs with AUC >0.600 are shown. Misclassification rate = (false positives + false negatives)/total sample counts, determined at the cut-off given. For abbreviations, see Table 1 and Table 2. Concentration ranges of BMs: ng/mg creatinine: ADP, AGT, ColXIII, HABP2, C4BP, CFH, CFI, FGG, GS, LRGP1, VTN, C1q; pg/mg creatinine: CRP, C9, FMN, PICP, TGF-β. Products of BM concentrations were built as in Figure 4.

BM	Cut-off	Sensitivity	1-Specificity	AUC	95%—Confidence Interval	Misclassification Rate	BM × BM	Cut-off	Sensitivity	1-Specificity	AUC	95%—Confidence Interval	Misclassification Rate
ColXIII	0.07	0.857	0.36	0.717	0.624–0.810	0.34	ADP × PICP	260.4	0.500	0.061	0.727	0.536–0.918	0.11
HABP2	0.29	0.636	0.194	0.744	0.595–0.894	0.21	AGT × GS	115.3	0.500	0.146	0.754	0.592–0.916	0.48
C4BP	3.46	0.692	0.240	0.693	0.512–0.874	0.27	AGT × C9	245.8	0.600	0.073	0.754	0.571–0.938	0.10
CRP	7.54	0.529	0.149	0.656	0.495–0.818	0.20	C4BP × ColXIII	0.53	0.667	0.235	0.750	0.603–0.897	0.27
CFH	8.89	0.667	0.313	0.697	0.542–0.853	0.31	GS × CFH	624.0	0.700	0.253	0.754	0.589–0.918	0.26
CFI	1.70	0.462	0.131	0.669	0.496–0.842	0.14	GS × C9	19728	0.600	0.091	0.754	0.581–0.926	0.12
							GS × CFI	184.1	0.700	0.111	0.796	0.635–0.958	0.13
							HABP2 × TGFβ	3.28	0.700	0.133	0.755	0.569–0.941	0.15
							HABP2 × FGG	365.4	0.700	0.092	0.812	0.678–0.947	0.11
							VTN × CFI	4.83	0.667	0.101	0.765	0.576–0.954	0.07
							CFH × CFI	27.3	0.636	0.152	0.762	0.599–0.925	0.17
							CFI × FGG	2883	0.583	0.040	0.755	0.586–0.924	0.08
							CFH × FGG	9180	0.727	0.232	0.771	0.610–0.931	0.14
							C4BP × ColXIII × HABP2	0.11	0.700	0.156	0.784	0.626–0.942	0.17

**Table 4 ijms-25-09320-t004:** Basic and clinical characteristics in the study population and controls.Data are presented as mean, (SD), and range (age, BMI). ^a^ *p* < 0.05 is considered significant, ^b^ *p* < 0.01 highly significant, Mann-Whitney U-test. Unmarked values indicate where patients are not significantly different from healthy people or females are not significantly different from male controls. We compared BMI and blood pressure in patients with healthy cohorts of the same age, sex and height and SDS transformed BMI [55] and blood pressure [56] to a z-score (cf. Appendix A). ^a^ and ^b^ indicate significant and highly significant differences between patients and controls. Healthy pediatric controls included 70 samples from the Leipzig Medical Biobank (Leipzig Research Center for Civilization Diseases. LIFE-child (collection: 02/2014-11/2014)) as well as 34 pediatric controls acquired in Jena (09/2020-09/2021).

	Group	n	Gender	Age [y]	BMI [SDS, z]	eGFR [mL/min/1.73m^2^]	Cystatin cGFR [mL/min/1.73m^2^]	Cystatin C [serum] mg/L	Blood Glucose [mmol/L]	HBA1c [%]	Blood Pressure Syst. [SDS, z]	Blood Pressure Diast [SDS, z]	Serum Creatinine [µmol/L]	Urine Albumin [mg/g creatinine]
	Total controls	104	48 m/56f	10.3 (4.0, 3.1–17.1)	0.069 (1.264, −3.44–3.68)	149.8 (28.2)	117.7 (23.0)	0.84 (0.12)	5.01 (1.44)	5.18 (0.28)	0.404 (0.999)	0.344 (0.943)	48.1 (14.4)	9.2 (9.9)
	Female controls	56	f	10.3 (4.0, 3.5–16.7	−0.064 (1.25, −3.44–2.90)	153.4 (30.0)	121.5 (22.5)	0.81 (0.11)	5.00 (1.88)	5.17 (0.28)	0.325 (1.00)	0.409 (1.01)	45.5 (12.7)	11.1 (12.0)
	Male controls	48	m	10.3 (4.1, 3.1–17.1)	0.220 (1.28, −2.50–3.68)	145.6 (25.6)	113.2 (23.1)	0.87 (0.12)	5.01 (0.60)	5.19 (0.29)	0.490 (0.997)	0.274 (0.866)	51.0 (15.8)	6.9 (5.9)
A	Metabolic syndrome, hypertension, obesity, T2DM	18	7 m/11 f	14.6 ^b^ (2.8)	2.55 ^b^ (0.85, 0.58–4.34)	154.8 (34.7)	112.1 (20.1)	0.83 (0.13)	6.01 ^b^ (1.29)	5.50 ^b^ (0.39)	2.048 ^b^ (1.27)	1.981 ^b^ (1.31)	53.2 (10.2)	5.3 (4.8)
B	T1DM	52	27 m/25 f	14.7 ^b^ (4.1)	0.629 ^b^ (1.16, −2.96–3.23)	n. d.	107.5 ^a^ (19.3)	0.87 ^a^ (0.14)	8.6 ^b^ (3.8)	7.68 ^b^ (1.01)	1.370 ^b^ (1.10)	1.456 ^b^ (0.932)	52.0 (19.0)	16.4 (49.5)
C	Renal agenesis, hypoplasia, multicystic renal dysplasia	33	21 m/12 f	11.4 (4.3)	0.268 (1.197, −2.68–3.25)	138.9 (27.4)	101.3 ^a^ (14.7)	0.94 ^b^ (0.11)	5.44 ^b^ (0.68)	5.27 (0.27)	1.452 ^b^ (1.27)	0.556 (0.837)	54.4 (15.5)	23.6 (47.3)
D	Anomalies of the urinary tract, duplex kidneys	19	11 m/8 f	10.9 (3.8)	0.438 (1.212, −2.68–3.25)	141.5 (27.9)	118.1 (28.7)	0.85 (0.16)	5.25^b^ (0.48)	5.23 (0.25)	1.647^b^ (0.856)	0.577 (0.815)	52.8 (14.1)	17.9 (24.5)
E	ADPKD, ARPKD	17	9 m/8 f	10.4 (4.5)	0.091 (0.930, −1.58–1.92)	156.7 (29.0)	117.1 (19.0)	0.84 (0.12)	5.25 ^a^ (0.27)	5.19 (0.25)	1.587 ^b^ (1.14)	1.101 (0.975)	45.9 (11.1)	95.6 ^b^ (205.5)
F	Nephron-ophthisis	3	1 m/2 f	8.5 (1.6)	−0.270 (0.488, −0.96–0.10)	137.8 (32.8)	99.3 (12.7)	0.98 ^a^ (0.11)	5.20 (0.43)	5.33 (0.25)	1.707 ^a^ (0.684)	1.771 ^b^ (0.441)	46.7 (12.5)	11.6 (5.7)
G	Post-infectious GN, IgAN, IgAV	9	3m/6 f	13.7 ^b^ (2.3)	0.083 (1.746, −2.31–2.43)	146.8 (29.7)	100.8 ^b^ (18.7)	0.93 ^b^ (0.14)	5.87 ^a^ (0.85)	5.34 (0.35)	1.001 (0.835)	0.054 (0.754)	55.0 (14.6)	57.6 ^a^ (65.3)

## Data Availability

All data are available in the main text or the Appendix A. Detailed anonymized clinical data that support the findings of this study are available on reasonable request. For general comments or other requests, please contact H.R.

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
