# Peer review of "Preclinical Detection of Early Glomerular Injury in Children with Kidney Diseases—Independently of Usual Markers of Kidney Impairment and Inflammation"

_ijms, 2024, doi:10.3390/ijms25179320_

Round 1

Reviewer 1 Report

Comments and Suggestions for Authors

Authors had made a novel approach to evaluate the protein biomarkers in various disease conditions and their purposes is to identify the pre-clinical disease for early intervention. Their intention is worth encouraging and the works were well performed. Comments are as the followings.

1. The Introduction is too lengthy, including 49 references. In my opinion, the study purpose is not difficult to explain. Authors can consider making it more concise.

2. As mentioned, the number of the test subjects were not enough, and the size was not evenly distributed. Of course, some diseased subjects are not easy to find and get together. Readers will sincerely hope that authors might amend the deficit.

3. Authors had proposed that several BMS, alone or in combination, might have reliable power to distinguish disease from healthy. However, the sensitivity, specificity, and AUC, were not high. Therefore, the real application of these BMs may be doubtful.

4. Is it possible that authors actually had follow-up data for these tested subjects? We know that there is no specific treatment for some of diseases in these patients. If they were identified by this method, how are they now? Do they benefit in any way from this screening?

Comments on the Quality of English Language

The English is mostly fine and understandable, but please have it read by someone who is familiar with native English. Just one example, in the very last paragraph of your Discussion, the sentence "Although some serum parameters discriminate between patients and controls, urine, noninvasive, is clinically preferable for practical reasons." is unclear to understand. Please read your text carefully and find those similar sentences, and edit them. 

Author Response

Reviewer 1

Authors had made a novel approach to evaluate the protein biomarkers in various disease conditions and their purposes is to identify the pre-clinical disease for early intervention. Their intention is worth encouraging and the works were well performed. Comments are as the followings.

Response: Thank you for the review and your comments helping us to improve our manuscript further. We provide a point by point response to your comments and we believe that we were able to address all of your comments.

  1. The Introduction is too lengthy, including 49 references. In my opinion, the study purpose is not difficult to explain. Authors can consider making it more concise.

Response: Thank you. We agree and shortened the introduction by one third from ~850 words to ~570 words in order to be more concise.

  1. As mentioned, the number of the test subjects were not enough, and the size was not evenly distributed. Of course, some diseased subjects are not easy to find and get together. Readers will sincerely hope that authors might amend the deficit.

Response: We agree. Indeed, it took us more than 10 years from IRB-approval, informed consent, and animal experiments until sample collection and final analysis/interpretation of the data. Aggravating, the sampling of human specimens was performed during the Corona pandemic, and hampered by the low frequency of on-site visits at this time.

As a response to the reviewer, we now mention this limitation in the discussion 5th paragraph on page 13.

  1. Authors had proposed that several BMS, alone or in combination, might have reliable power to distinguish disease from healthy. However, the sensitivity, specificity, and AUC, were not high. Therefore, the real application of these BMs may be doubtful.

Response: We agree with the reviewer that „real world“ application of our BMs still is doubtful because of our limited sample size. We are aware of a further, still existing limitation of our study, the unknown time courses of our BMs. Thus, the higher heterogeneity within our patient´s groups probably resulted in lower sensitivity, specificity, and AUC compared to the clearer results of the more homogeneous groups of AS-patients [49]. However, our biomarker study is at least a serious attempt to achieve meaningful results with commercially available ELISAs that are reproducible under "real world" conditions.

  1. Is it possible that authors actually had follow-up data for these tested subjects? We know that there is no specific treatment for some of diseases in these patients. If they were identified by this method, how are they now? Do they benefit in any way from this screening?

Response: Thank you for the comment. Yes, indeed, the ultimate goal of our biomarker attempt was to identify children at early stages of CKD – which might translate to a benefit (delay of progression of CKD) and to open the opportunity to develop and test new therapy options at early stages. You are right, the follow-up regarding outcome must follow. Indeed, this is already “work in progress”, as well as the analysis of time courses of our BMs.

As a response to the reviewer’s comment we added the new sentence to the last paragraph of discussion about BM as screening method and its potential benefits.  

Comments on the Quality of English Language

The English is mostly fine and understandable, but please have it read by someone who is familiar with native English. Just one example, in the very last paragraph of your Discussion, the sentence "Although some serum parameters discriminate between patients and controls, urine, noninvasive, is clinically preferable for practical reasons." is unclear to understand. Please read your text carefully and find those similar sentences, and edit them.

Response: Thank you for this comment, well taken. A native speaker (Dr. A. Davis), also familiar with scientific writing, had improved the language. We tried to adjust “difficult” sentences into two separate sentences in order to make them easier to understand.

Thank you very much for your consideration,

Heidrun Rhode and Oliver Gross (on behalf of all other authors)

Reviewer 2

The manuscript “Preclinical detection of early glomerular injury in children with kidney diseases - independently of usual markers of kidney impairment and inflammation”, represents a continuation of a previously published work in mouse models of Alport syndrome and Alport syndrome patients. The previous study identified specific biomarkers (BMs) reflecting glomerular dysfunction associated with Alport Syndrome. The current manuscript by Rhode at al describes screening for the same BMs in several distinct types of kidney disease. The results are interesting and copious, the statistical analysis is appropriate. However, I feel that the study structure needs some changes to improve the flow and to emphasize the findings appropriately. For example, the Introduction has a disjointed description of various forms of CKD without clarity why the authors chose to screen these patients. Surprisingly, the Introduction section has only two sentences about the previous Alport syndrome study that is critical for the new paper. Furthermore, Figure 1, as well as some text in the Results section, are somewhat misleading since they describe some methods and results that have already been published but are not part of the current manuscript. With some re-arranging and re-writing of the text, however, the manuscript could be streamlined and improved.

Response: Thank you for the review and recommendations. As a response to your comment on the introduction (see also recommendations reviewer 1), we have shortened the introduction in order to be more concise with the rationale of the study. We understand your comment on Alport syndrome, but we want to avoid the impression that we publish duplicates. As a further response to your comment, now we added further information about our previous publication in the legend of Figure 1. As most biomarkers were measured in relation to our (previous published) data in children with Alport syndrome, it is difficult to make a clear cut between some of our data from [49] and the data of the present study shown in the results section. According to your recommendation and your comment #1, we now clarify the structure of our biomarker-studies and added in the first paragraph of the results section: “In our previous studies [49,50], mammalian models of Alport Syndrome and children with Alport Syndrome were screened initially to identify generic markers for glomerular disease. In our current study, these markers are now being tested in patients with various kidney diseases (including CAKUT, cystic disease and diabetes) to see whether they represent useful BMs for other forms of kidney disease (Figure 1).” 

Comments:

  1. Both abstract and Introduction are a bit convoluted making it difficult to figure our which disease(s) are being discussed and why. From the Figure 1, it becomes clear that mammalian models of Alport Syndrome and children with Alport Syndrome were screened initially to identify generic markers for glomerular disease. However, these markers are now being tested in patients with various kidney diseases (including CAKUT, cystic disease and diabetes) to see whether they represent useful BMs for other forms of kidney disease. Perhaps, it would be useful to articulate this in the Introduction. The authors should offer an interpretation of how these BMs should be used - as selective markers of glomerular injury or of any form of kidney injury and why.

Response: Agree, thank you for your help. See our response above:  we now clarify the structure of our biomarker-studies and added in the first paragraph of the results section: “In our previous studies [49,50], mammalian models of Alport Syndrome and children with Alport Syndrome were screened initially to identify generic markers for glomerular disease. In our current study, these markers are now being tested in patients with various kidney diseases (including CAKUT, cystic disease and diabetes) to see whether they represent useful BMs for other forms of kidney disease (Figure 1).” 

  1. The “discovery” part of the project has been published and this should be clearly stated. In the beginning of the Results section, the authors should remove “discovery” and “proof-of-principal” wording – these descriptors could be moved to the Introduction, to set the stage for the Results shown in the Fig2 and below. Figure 1 should be changed accordingly to show only what belongs to the current paper. As is, this Figure 1 is somewhat misleading since the reader expects to see the Alport syndrome studies.

Response: In response to your recommendation, we now clearly state in the first sentence of the result section that the discovery part has been published, but now builds the basis of our proof-of-concept phase. We feel that this clarification is better placed as “introduction” of the results section (rather than as part of the introduction), because it might be easier to understand our rationale. As further response, we changed figure 1 and now clarify that the focus is on the “proof-of-concept phase”, which is indicated by a red frame. We hope that the figure is more informative and less misleading now.

  1. Lane 100 – Alport syndrome is known in “all mammals”. The citation refers to dogs and mice. Thus, it would be more appropriate to specify just dogs and transgenic mice in which Alport syndrome was reported/studied. In the same paragraph, the authors write ”mammal species” and “mammal models”; it should be “mammalian models”. Please correct.

Response: Thank you very much, we agree. We already changed this according to reviewer 1. According to your comment (and reviewer 1) changed to “Type IV collagen is preserved in all mammalian species. Therefore, it was possible to use Alport mammalian models to study pathogenesis and therapy [48], to study glomerular injury, and to search for early BMs [49, 50].

  1. Figure 2: I am not sure why there is a division between “C” (unilateral renal agenesis, solitary kidney, unilateral hypoplasia, dysplastic cystic kidney) and “D” (CAKUT). It seems that “C” is part of CAKUT and should be represented as such. Did the authors mean to identify the markers associated specifically with solitary kidney? If so, this needs to be mentioned in the results and the Discussion. The justification for why CAKUT patients are included to be screened for glomerular BMs is also unclear.

Response: We understand your question, but – according to our pediatric nephrologists – we did want to distinguish between intrarenal dysplasia and unilateral stress (C) directly impairing kidney function, and urogenital malformation without, primary, kidney impairment, but leading to reflux and tubular fibrosis, and secondarily affecting kidney function (D).

  1. Figure 2. Please indicate the number of patients for each type of disease on the diagram.

Response: Numbers of patients are now added to figure 2.

  1. Figure 4. Please elaborate in more detail how combined values for several BMs were calculated.

Response: In urine all BM concentrations (c; concentration/ml) were normalized by the respective urinary creatinine concentration (mg/ml) as given in Figure 3. Throughout, only these normalized values are used and compared. Please see Table 1 for concentration ranges. Thus, in Figure 4 the products of BMs represent the products of their normalized concentrations. Here – and in Tables 2 and 3, and in Figure 5 - the measures (units) exactly would be cBM1 x cBM2/(mg creatinine)2 or  cBM1 x cBM2 x cBM3/(mg creatinine)3. For simplification, only BM1 x BM2 or BM1 x BM2 x BM3 was used on the y-axes of Figures 4 and 5 and in Tables 2 and 3. A short description is now provided in the Figure legends.

  1. The discussion part is somewhat disjointed and missing a unifying statement about what was discovered in the current manuscript. Thus, it would be appropriate to write a succent paragraph to state the major finding of the paper. Interestingly, the authors screened some patients with glomerular diseases (group G: post-infectious GN, IgAN, IgAV): as expected, all BMs identified in the Alport syndrome were clearly elevated in these patients. This is a strong validation of the biomarkers and the methods used and, perhaps, should be stressed at the start of the Discussion – or organized separately in the Results.

Response: Thank you very much. We included such a statement within the discussion section now.

  1. Another aspect that should be stressed in the Discussion is the great variability of the BM concentrations in both controls and patients. It is a bit baffling to see 1000 – 100000 fold increase in various samples (from 0.01ng/ml to 1000ng/ml). Such variability questions the validity of the work. Please elaborate.

Response: Thank you very much for the comment. You are right.

Please compare the variability of singular parameters (Fig. 3) and those of combined parameters. Apart from values below the detection limit (zero, more abundant in the control group), the variability of singular parameters is in the range of one or two orders of magnitude, similar to many clinical chemical parameters. And, their medians differ by about one order of magnitude. In contrast the variability of combined (multiplied) concentrations are expectedly much higher, since two or three parameters were multiplied. However, despite high variability and some overlapping values the significance values of these combined values are higher than that of singular values.

Nevertheless, we are aware that our findings have to be further validated in much greater cohorts of patients as well as in studies on their outcome. The latter is already “work in progress”.

Thank you very much for your consideration,

Heidrun Rhode and Oliver Gross (on behalf of all other authors)

Reviewer 2 Report

Comments and Suggestions for Authors

The manuscript “Preclinical detection of early glomerular injury in children with kidney diseases - independently of usual markers of kidney impairment and inflammation”, represents a continuation of a previously published work in mouse models of Alport syndrome and Alport syndrome patients. The previous study identified specific biomarkers (BMs) reflecting glomerular dysfunction associated with Alport Syndrome. The current manuscript by Rhode at al describes screening for the same BMs in several distinct types of kidney disease. The results are interesting and copious, the statistical analysis is appropriate. However, I feel that the study structure needs some changes to improve the flow and to emphasize the findings appropriately. For example, the Introduction has a disjointed description of various forms of CKD without clarity why the authors chose to screen these patients. Surprisingly, the Introduction section has only two sentences about the previous Alport syndrome study that is critical for the new paper. Furthermore, Figure 1, as well as some text in the Results section, are somewhat misleading since they describe some methods and results that have already been published but are not part of the current manuscript. With some re-arranging and re-writing of the text, however, the manuscript could be streamlined and improved.

Comments:

Both abstract and Introduction are a bit convoluted making it difficult to figure our which disease(s) are being discussed and why.  From the Figure 1, it becomes clear that mammalian models of Alport Syndrome and children with Alport Syndrome were screened initially to identify generic markers for glomerular disease. However, these markers are now being tested in patients with various kidney diseases (including CAKUT, cystic disease and diabetes) to see whether they represent useful BMs for other forms of kidney disease. Perhaps, it would be useful to articulate this in the Introduction. The authors should offer an interpretation of how these BMs should be used - as selective markers of glomerular injury or of any form of kidney injury and why.

The “discovery” part of the project has been published and this should be clearly stated. In the beginning of the Results section, the authors should remove “discovery” and “proof-of-principal” wording – these descriptors could be moved to the Introduction, to set the stage for the Results shown in the Fig2 and below. Figure 1 should be changed accordingly to show only what belongs to the current paper. As is, this Figure 1 is somewhat misleading since the reader expects to see the Alport syndrome studies.

Lane 100 – Alport syndrome is known in “all mammals”. The citation refers to dogs and mice.  Thus, it would be more appropriate to specify just dogs and transgenic mice in which Alport syndrome was reported/studied. In the same paragraph, the authors write ”mammal species” and “mammal models”; it should be “mammalian models”. Please correct.

Figure 2: I am not sure why there is a division between “C” (unilateral renal agenesis, solitary kidney, unilateral hypoplasia, dysplastic cystic kidney) and “D” (CAKUT). It seems that “C” is part of CAKUT and should be represented as such. Did the authors mean to identify the markers associated specifically with solitary kidney? If so, this needs to be mentioned in the results and the Discussion. The justification for why CAKUT patients are included to be screened for glomerular BMs is also unclear.

Figure 2. Please indicate the number of patients for each type of disease on the diagram.

Figure 4. Please elaborate in more detail how combined values for several BMs were calculated.

The discussion part is somewhat disjointed and missing a unifying statement about what was discovered in the current manuscript. Thus, it would be appropriate to write a succent paragraph to state the major finding of the paper. Interestingly, the authors screened some patients with glomerular diseases (group G: post-infectious GN, IgAN, IgAV): as expected, all BMs identified in the Alport syndrome were clearly elevated in these patients. This is a strong validation of the biomarkers and the methods used and, perhaps, should be stressed at the start of the Discussion – or organized separately in the Results.

Another aspect that should be stressed in the Discussion is the great variability of the BM concentrations in both controls and patients. It is a bit baffling to see 1000 – 100000 fold increase in various samples (from 0.01ng/ml to 1000ng/ml). Such variability questions the validity of the work. Please elaborate.

Author Response

Reviewer 2

The manuscript “Preclinical detection of early glomerular injury in children with kidney diseases - independently of usual markers of kidney impairment and inflammation”, represents a continuation of a previously published work in mouse models of Alport syndrome and Alport syndrome patients. The previous study identified specific biomarkers (BMs) reflecting glomerular dysfunction associated with Alport Syndrome. The current manuscript by Rhode at al describes screening for the same BMs in several distinct types of kidney disease. The results are interesting and copious, the statistical analysis is appropriate. However, I feel that the study structure needs some changes to improve the flow and to emphasize the findings appropriately. For example, the Introduction has a disjointed description of various forms of CKD without clarity why the authors chose to screen these patients. Surprisingly, the Introduction section has only two sentences about the previous Alport syndrome study that is critical for the new paper. Furthermore, Figure 1, as well as some text in the Results section, are somewhat misleading since they describe some methods and results that have already been published but are not part of the current manuscript. With some re-arranging and re-writing of the text, however, the manuscript could be streamlined and improved.

Response: Thank you for the review and recommendations. As a response to your comment on the introduction (see also recommendations reviewer 1), we have shortened the introduction in order to be more concise with the rationale of the study. We understand your comment on Alport syndrome, but we want to avoid the impression that we publish duplicates. As a further response to your comment, now we added further information about our previous publication in the legend of Figure 1. As most biomarkers were measured in relation to our (previous published) data in children with Alport syndrome, it is difficult to make a clear cut between some of our data from [49] and the data of the present study shown in the results section. According to your recommendation and your comment #1, we now clarify the structure of our biomarker-studies and added in the first paragraph of the results section: “In our previous studies [49,50], mammalian models of Alport Syndrome and children with Alport Syndrome were screened initially to identify generic markers for glomerular disease. In our current study, these markers are now being tested in patients with various kidney diseases (including CAKUT, cystic disease and diabetes) to see whether they represent useful BMs for other forms of kidney disease (Figure 1).” 

Comments:

  1. Both abstract and Introduction are a bit convoluted making it difficult to figure our which disease(s) are being discussed and why. From the Figure 1, it becomes clear that mammalian models of Alport Syndrome and children with Alport Syndrome were screened initially to identify generic markers for glomerular disease. However, these markers are now being tested in patients with various kidney diseases (including CAKUT, cystic disease and diabetes) to see whether they represent useful BMs for other forms of kidney disease. Perhaps, it would be useful to articulate this in the Introduction. The authors should offer an interpretation of how these BMs should be used - as selective markers of glomerular injury or of any form of kidney injury and why.

Response: Agree, thank you for your help. See our response above:  we now clarify the structure of our biomarker-studies and added in the first paragraph of the results section: “In our previous studies [49,50], mammalian models of Alport Syndrome and children with Alport Syndrome were screened initially to identify generic markers for glomerular disease. In our current study, these markers are now being tested in patients with various kidney diseases (including CAKUT, cystic disease and diabetes) to see whether they represent useful BMs for other forms of kidney disease (Figure 1).” 

  1. The “discovery” part of the project has been published and this should be clearly stated. In the beginning of the Results section, the authors should remove “discovery” and “proof-of-principal” wording – these descriptors could be moved to the Introduction, to set the stage for the Results shown in the Fig2 and below. Figure 1 should be changed accordingly to show only what belongs to the current paper. As is, this Figure 1 is somewhat misleading since the reader expects to see the Alport syndrome studies.

Response: In response to your recommendation, we now clearly state in the first sentence of the result section that the discovery part has been published, but now builds the basis of our proof-of-concept phase. We feel that this clarification is better placed as “introduction” of the results section (rather than as part of the introduction), because it might be easier to understand our rationale. As further response, we changed figure 1 and now clarify that the focus is on the “proof-of-concept phase”, which is indicated by a red frame. We hope that the figure is more informative and less misleading now.

  1. Lane 100 – Alport syndrome is known in “all mammals”. The citation refers to dogs and mice. Thus, it would be more appropriate to specify just dogs and transgenic mice in which Alport syndrome was reported/studied. In the same paragraph, the authors write ”mammal species” and “mammal models”; it should be “mammalian models”. Please correct.

Response: Thank you very much, we agree. We already changed this according to reviewer 1. According to your comment (and reviewer 1) changed to “Type IV collagen is preserved in all mammalian species. Therefore, it was possible to use Alport mammalian models to study pathogenesis and therapy [48], to study glomerular injury, and to search for early BMs [49, 50].

  1. Figure 2: I am not sure why there is a division between “C” (unilateral renal agenesis, solitary kidney, unilateral hypoplasia, dysplastic cystic kidney) and “D” (CAKUT). It seems that “C” is part of CAKUT and should be represented as such. Did the authors mean to identify the markers associated specifically with solitary kidney? If so, this needs to be mentioned in the results and the Discussion. The justification for why CAKUT patients are included to be screened for glomerular BMs is also unclear.

Response: We understand your question, but – according to our pediatric nephrologists – we did want to distinguish between intrarenal dysplasia and unilateral stress (C) directly impairing kidney function, and urogenital malformation without, primary, kidney impairment, but leading to reflux and tubular fibrosis, and secondarily affecting kidney function (D).

  1. Figure 2. Please indicate the number of patients for each type of disease on the diagram.

Response: Numbers of patients are now added to figure 2.

  1. Figure 4. Please elaborate in more detail how combined values for several BMs were calculated.

Response: In urine all BM concentrations (c; concentration/ml) were normalized by the respective urinary creatinine concentration (mg/ml) as given in Figure 3. Throughout, only these normalized values are used and compared. Please see Table 1 for concentration ranges. Thus, in Figure 4 the products of BMs represent the products of their normalized concentrations. Here – and in Tables 2 and 3, and in Figure 5 - the measures (units) exactly would be cBM1 x cBM2/(mg creatinine)2 or  cBM1 x cBM2 x cBM3/(mg creatinine)3. For simplification, only BM1 x BM2 or BM1 x BM2 x BM3 was used on the y-axes of Figures 4 and 5 and in Tables 2 and 3. A short description is now provided in the Figure legends.

  1. The discussion part is somewhat disjointed and missing a unifying statement about what was discovered in the current manuscript. Thus, it would be appropriate to write a succent paragraph to state the major finding of the paper. Interestingly, the authors screened some patients with glomerular diseases (group G: post-infectious GN, IgAN, IgAV): as expected, all BMs identified in the Alport syndrome were clearly elevated in these patients. This is a strong validation of the biomarkers and the methods used and, perhaps, should be stressed at the start of the Discussion – or organized separately in the Results.

Response: Thank you very much. We included such a statement within the discussion section now.

  1. Another aspect that should be stressed in the Discussion is the great variability of the BM concentrations in both controls and patients. It is a bit baffling to see 1000 – 100000 fold increase in various samples (from 0.01ng/ml to 1000ng/ml). Such variability questions the validity of the work. Please elaborate.

Response: Thank you very much for the comment. You are right.

Please compare the variability of singular parameters (Fig. 3) and those of combined parameters. Apart from values below the detection limit (zero, more abundant in the control group), the variability of singular parameters is in the range of one or two orders of magnitude, similar to many clinical chemical parameters. And, their medians differ by about one order of magnitude. In contrast the variability of combined (multiplied) concentrations are expectedly much higher, since two or three parameters were multiplied. However, despite high variability and some overlapping values the significance values of these combined values are higher than that of singular values.

Nevertheless, we are aware that our findings have to be further validated in much greater cohorts of patients as well as in studies on their outcome. The latter is already “work in progress”.

Thank you very much for your consideration,

Heidrun Rhode and Oliver Gross (on behalf of all other authors)

Reviewer 1

Authors had made a novel approach to evaluate the protein biomarkers in various disease conditions and their purposes is to identify the pre-clinical disease for early intervention. Their intention is worth encouraging and the works were well performed. Comments are as the followings.

Response: Thank you for the review and your comments helping us to improve our manuscript further. We provide a point by point response to your comments and we believe that we were able to address all of your comments.

  1. The Introduction is too lengthy, including 49 references. In my opinion, the study purpose is not difficult to explain. Authors can consider making it more concise.

Response: Thank you. We agree and shortened the introduction by one third from ~850 words to ~570 words in order to be more concise.

  1. As mentioned, the number of the test subjects were not enough, and the size was not evenly distributed. Of course, some diseased subjects are not easy to find and get together. Readers will sincerely hope that authors might amend the deficit.

Response: We agree. Indeed, it took us more than 10 years from IRB-approval, informed consent, and animal experiments until sample collection and final analysis/interpretation of the data. Aggravating, the sampling of human specimens was performed during the Corona pandemic, and hampered by the low frequency of on-site visits at this time.

As a response to the reviewer, we now mention this limitation in the discussion 5th paragraph on page 13.

  1. Authors had proposed that several BMS, alone or in combination, might have reliable power to distinguish disease from healthy. However, the sensitivity, specificity, and AUC, were not high. Therefore, the real application of these BMs may be doubtful.

Response: We agree with the reviewer that „real world“ application of our BMs still is doubtful because of our limited sample size. We are aware of a further, still existing limitation of our study, the unknown time courses of our BMs. Thus, the higher heterogeneity within our patient´s groups probably resulted in lower sensitivity, specificity, and AUC compared to the clearer results of the more homogeneous groups of AS-patients [49]. However, our biomarker study is at least a serious attempt to achieve meaningful results with commercially available ELISAs that are reproducible under "real world" conditions.

  1. Is it possible that authors actually had follow-up data for these tested subjects? We know that there is no specific treatment for some of diseases in these patients. If they were identified by this method, how are they now? Do they benefit in any way from this screening?

Response: Thank you for the comment. Yes, indeed, the ultimate goal of our biomarker attempt was to identify children at early stages of CKD – which might translate to a benefit (delay of progression of CKD) and to open the opportunity to develop and test new therapy options at early stages. You are right, the follow-up regarding outcome must follow. Indeed, this is already “work in progress”, as well as the analysis of time courses of our BMs.

As a response to the reviewer’s comment we added the new sentence to the last paragraph of discussion about BM as screening method and its potential benefits.  

Comments on the Quality of English Language

The English is mostly fine and understandable, but please have it read by someone who is familiar with native English. Just one example, in the very last paragraph of your Discussion, the sentence "Although some serum parameters discriminate between patients and controls, urine, noninvasive, is clinically preferable for practical reasons." is unclear to understand. Please read your text carefully and find those similar sentences, and edit them.

Response: Thank you for this comment, well taken. A native speaker (Dr. A. Davis), also familiar with scientific writing, had improved the language. We tried to adjust “difficult” sentences into two separate sentences in order to make them easier to understand.

Thank you very much for your consideration,

Heidrun Rhode and Oliver Gross (on behalf of all other authors)

Round 2

Reviewer 2 Report

Comments and Suggestions for Authors

The authors have reworked the manuscript, particularly, the Introduction and the beginning of the Result sections. This certainly improved the paper and defined the scope of this work more precisely.  I still feel, however, that the Discussion needs improvement and consolidation. In many instances, the authors repeat the results and follow every permutation of the changes for multiple markers for various diseases, making the text difficult to follow and leaving no clear message for the readers. I would suggest to start the Discussion with the generalized statement of the major finding and chose several of the major findings to discuss.

Definition of Groups C and D is still rather unusual since the entities described in both groups usually are included into CAKUT. It is true that CAKUT phenotypes may arise as a consequences of genetic and  environmental inputs, or an interplay of both.  Perhaps, the authors could name both groups as CAKUT, but present them as CAKUT-C and CAKUT-D.   

Author Response

Reviewer 2:

"The authors have reworked the manuscript, particularly, the Introduction and the beginning of the Result sections. This certainly improved the paper and defined the scope of this work more precisely.
1) I still feel, however, that the Discussion needs improvement and consolidation. In many instances, the authors repeat the results and follow every permutation of the changes for multiple markers for various diseases, making the text difficult to follow and leaving no clear message for the readers. I would suggest to start the Discussion with the generalized statement of the major finding and chose several of the major findings to discuss.

2) Definition of Groups C and D is still rather unusual since the entities described in both groups usually are included into CAKUT. It is true that CAKUT phenotypes may arise as a consequences of genetic and environmental inputs, or an interplay of both. Perhaps, the authors could name both groups as CAKUT, but present them as CAKUT-C and CAKUT-D."

Response: Thank you for your feedback. You are absolutely right, and we agree, that both groups, C and D, cover congenital disorders. Of course, renal agenesis and, for example, MCKD (multicystic kidney dysplasia) are forms of congenital urinary tract anomalies. Thus, you are right, solitary functioning kidney (SFK) might be categorized as a subgroup of the CAKUT, which was proposed in Congenital Solitary Functioning Kidney: A Review. Wakabayashi EA et al. Curr Med Chem. 2023;30(2):203-219. doi: 10.2174/0929867329666220629142556.PMID: 35770397. 

However, in clinical practice in pediatric nephrology in our University hospitals, congenital anomalies are actually divided in the different groups of diseases (such as C and D) with respect to different functional implications and therapeutic options.

Multi-cystic dysplasia, unilateral hypoplasia, and solitary kidney (C), represent one – above mentioned - disease group with one functioning solitary kidney (or solitary functioning kidney) leading to hyperfiltration and, thus to glomerular impairment of this single kidney, followed by proteinuria. (Clinical implications of the solitary functioning kidney. Westland Ret al. Clin J Am Soc Nephrol. 2014 May;9(5):978-86. doi: 10.2215/CJN.08900813. Epub 2013 Dec 26.PMID: 24370773; Outcomes of solitary functioning kidneys-renal agenesis is different than multicystic dysplastic kidney disease.Matsell DG et al. Pediatr Nephrol. 2021 Nov;36(11):3673-3680. doi: 10.1007/s00467-021-05064-1. Epub 2021 May 5.PMID: 33954810) 

The term CAKUT – congenital abnormalities of the kidney and urinary tract - has been reserved for other anomalies for a long time, i.e. those of the entire urogenital tract, and, additionally, duplex kidney. (Nat Rev Nephrol. 2023 Nov;19(11):709-720. doi: 10.1038/s41581-023-00742-9. Epub 2023 Jul 31. The genetics and pathogenesis of CAKUT. CM Kolvenbach et al.). In these diseases, a higher intra-renal and post-renal pressure can be expected (due to reflux) within the otherwise primarily unaffected kidneys due to urine retainment/reflux within the urogenital tract. This might/will lead to kidney impairment too.

We formed these two groups (C and D) – and all others - due to such different functional implications to analyze, what alteration of our BMs follows different pathophysiology’s.

When focusing on pathophysiological mechanisms, conditions with glomerular hyperfiltration of the remaining functional kidney must be grouped in one group. This includes also, e.g., post-nephrectomy, living kidney donation, and nephroblastoma-kidneys into the category of functional solitary kidneys. This is necessary because, glomerular hyperfiltration and reduced urine outflow with tubulointerstitial damage, recurrent infections, and parenchymal scarring lead to a completely different pattern of intrarenal injury and consequently to an altered biomarker profile.

As a response to your first comment (discussion), we re-arranged the text of the discussion in lines 270 to 330 (see track changes in the manuscript).

As a response to your second comment (CAKUT group C and D), we changed the text within the manuscript, see lines 273 and 323, to clarify the analytical value of our group C and D formation. We understand your arguments for different subgroups that would/could/might be also reasonable. However, redefining groups C and D would lead to major new statistical evaluations and we ask for your understanding that the new denominations CAKUT-C and -D could confuse a main group of our readers, pediatric nephrologists.

Thank you very much for your consideration,

Oliver Gross and Heidrun Rhode (on behalf of the other authors)
